# Anti-CD44 Variant 10 Monoclonal Antibody Exerts Antitumor Activity in Mouse Xenograft Models of Oral Squamous Cell Carcinomas

**DOI:** 10.3390/ijms25179190

**Published:** 2024-08-24

**Authors:** Kenichiro Ishikawa, Hiroyuki Suzuki, Tomokazu Ohishi, Guanjie Li, Tomohiro Tanaka, Manabu Kawada, Akira Ohkoshi, Mika K. Kaneko, Yukio Katori, Yukinari Kato

**Affiliations:** 1Department of Antibody Drug Development, Tohoku University Graduate School of Medicine, 2-1 Seiryo-machi, Aoba-ku, Sendai 980-8575, Miyagi, Japan; ken.ishikawa.r3@dc.tohoku.ac.jp (K.I.); clownair716@gmail.com (G.L.); tomohiro.tanaka.b5@tohoku.ac.jp (T.T.); mika.kaneko.d4@tohoku.ac.jp (M.K.K.); 2Department of Otolaryngology, Head and Neck Surgery, Tohoku University Graduate School of Medicine, 1-1 Seiryo-machi, Aoba-ku, Sendai 980-8575, Miyagi, Japan; ohkoshia@hotmail.com (A.O.); yukio.katori.d1@tohoku.ac.jp (Y.K.); 3Institute of Microbial Chemistry (BIKAKEN), Numazu, Microbial Chemistry Research Foundation, 18-24 Miyamoto, Numazu-shi 410-0301, Shizuoka, Japan; ohishit@bikaken.or.jp; 4Institute of Microbial Chemistry (BIKAKEN), Laboratory of Oncology, Microbial Chemistry Research Foundation, 3-14-23 Kamiosaki, Shinagawa-ku, Tokyo 141-0021, Japan; kawadam@bikaken.or.jp

**Keywords:** monoclonal antibody therapy, CD44v10, ADCC, CDC, oral cancer

## Abstract

CD44 regulates cell adhesion, proliferation, survival, and stemness and has been considered a tumor therapy target. CD44 possesses the shortest CD44 standard (CD44s) and a variety of CD44 variant (CD44v) isoforms. Since the expression of CD44v is restricted in epithelial cells and carcinomas compared to CD44s, CD44v has been considered a promising target for monoclonal antibody (mAb) therapy. We previously developed an anti-CD44v10 mAb, C_44_Mab-18 (IgM, kappa), to recognize the variant exon 10-encoded region. In the present study, a mouse IgG_2a_ version of C_44_Mab-18 (C_44_Mab-18-mG_2a_) was generated to evaluate the antitumor activities against CD44-positive cells compared with the previously established anti-pan CD44 mAb, C_44_Mab-46-mG_2a_. C_44_Mab-18-mG_2a_ exhibited higher reactivity compared with C_44_Mab-46-mG_2a_ to CD44v3–10-overexpressed CHO-K1 (CHO/CD44v3–10) and oral squamous cell carcinoma cell lines (HSC-2 and SAS) in flow cytometry. C_44_Mab-18-mG_2a_ exerted a superior antibody-dependent cellular cytotoxicity (ADCC) against CHO/CD44v3–10. In contrast, C_44_Mab-46-mG_2a_ showed a superior complement-dependent cytotoxicity (CDC) against CHO/CD44v3–10. A similar tendency was observed in ADCC and CDC against HSC-2 and SAS. Furthermore, administering C_44_Mab-18-mG_2a_ or C_44_Mab-46-mG_2a_ significantly suppressed CHO/CD44v3–10, HSC-2, and SAS xenograft tumor growth compared with the control mouse IgG_2a_. These results indicate that C_44_Mab-18-mG_2a_ could be a promising therapeutic regimen for CD44v10-positive tumors.

## 1. Introduction

CD44 is involved in tumor malignant progression through the promotion of tumor cell proliferation, migration, invasiveness, and stemness [1,2]. The variety of CD44 function is mediated by the alternative splicing of 20 exons [3,4,5]. CD44 standard (CD44s), the shortest isoform of CD44, is generated by the first five (1–5) and the last five (16–20) exons and expressed in a broad range of tissues [6,7,8]. The central (6–15) exons are alternatively spliced and inserted between the first and last five exons of CD44s. The variant exon-containing CD44 is designated as the CD44 variant (CD44v) isoform [9,10,11,12].

The CD44 ectodomain includes a hyaluronic acid (HA)-binding domain (HABD) [13,14,15]. The HABD is present in both CD44s and CD44v isoforms. Upon HA binding, CD44s and CD44v transduce the intracellular signaling through the cytoplasmic domain, which promotes cell migration and proliferation [16]. The variant exon-encoding regions possess a variety of functions. The v3-encoded region can be attached to a heparan sulfate side chain, which recruits heparin-binding growth factors and stimulates the signal transduction through activation of the receptors [4,17,18]. The v6-encoded region potentiates the MET signaling pathway through the formation of a ternary complex with its ligand, hepatocyte growth factor [19,20,21,22]. Moreover, the v8–10-encoded region regulates the intracellular reduced glutathione levels through the promotion of a cystine–glutamate transporter function [23]. These functions are essential for tumor cell proliferation, invasiveness, and survival to oxidative stress and chemotherapeutic drugs [24,25,26,27,28,29]. Therefore, CD44 has been considered as an essential target for tumor therapy [14,30].

Monoclonal antibodies (mAbs) against CD44 have been evaluated in clinical trials [31,32]. RG7356, a humanized anti-pan-CD44 mAb, exhibited the antitumor effect for B cell leukemia but no cytotoxicity on normal B cells [33]. In a human chronic lymphocytic leukemia-engrafted mouse model, RG7356 administration resulted in complete clearance of engrafted leukemia cells [33]. In acute myeloid leukemia [34] and advanced CD44-positive solid tumors [35], phase I clinical trials were conducted. Although RG7356 exhibited an acceptable safety profile, the studies were terminated due to the lack of dose–response relationship with RG7356 in both clinical and pharmacodynamic aspects [35].

Since CD44v expression is restricted in epithelial tissue and carcinomas, anti-CD44v mAbs were developed and evaluated in clinical studies. Humanized anti-CD44v6 mAbs (BIWA-4 and BIWA-8) labeled with ^186^Re showed antitumor efficacy in head and neck squamous cell carcinoma (SCC) xenograft-bearing mice [36]. Moreover, the antibody–drug conjugate (ADC) of BIWA-4, bivatuzumab–mertansine was developed and evaluated in clinical trials [32]. However, the clinical trials were terminated due to severe toxicity in the skin, probably due to the efficient accumulation of mertansine in the skin [32,37]. Therefore, anti-CD44 mAbs with more potent efficacy and lower toxicity to normal cells are desired.

The Fc region of therapeutic mAb binds to FcγRs on dendritic cells, macrophages, and neutrophils, which influences the adaptive immune responses through antigen presentation and cytokine production [38]. Moreover, the FcγR binding results in the activation of natural killer (NK) cells [39] and macrophages [40], which mediates antibody-dependent cellular cytotoxicity (ADCC). The complement-dependent cellular cytotoxicity (CDC) is also considered as an essential effector function in tumor immunotherapy [41]. The Fc region of therapeutic mAbs binds to complement C1q, facilitating the assembly of active C1 complex (C1q, C1r, and C1s). The reaction of the complement cascade finally promotes the assembly of the pore-forming membrane attack complex (MAC or C5b–C9) on the tumor cell membrane, which results in the terminal cell lysis [41]. The involvement of CDC in the antitumor effect was first recognized in the treatment of B cell lymphomas by an anti-CD20 mAb, rituximab [42,43]. Furthermore, the cytolytic capacity by complement has been shown in anti-CD38 and CD52 immunotherapies for multiple myeloma and chronic lymphocytic leukemia, respectively [43,44,45]. Moreover, a growing body of evidence suggests that complement plays critical roles in not only tumor cell lysis but also in several immunologic functions in antitumor immunity [46,47]. In the immunotherapy against solid tumors, an anti-HER2 bispecific and biparatopic antibody zanidatamab exerted more potent CDC against HER2-positive breast cancers compared with clinically approved anti-HER2 mAb trastuzumab [48].

The Cell-Based Immunization and Screening method is a strategy to obtain mAbs against a membrane protein comprehensively. We immunized mice with the CD44v3–10-overexpressed cells or CD44v3–10 ectodomain and established various anti-CD44 mAbs and determined their epitopes. C_44_Mab-5 [49] and C_44_Mab-46 [50] are anti-pan-CD44 mAbs, which have the epitopes within the constant exon 2- and 5-encoded sequences, respectively [51,52]. We also established various CD44v-specific mAbs. C_44_Mab-6 recognizes variant exon 3-encoded sequences and is referred to as anti-CD44v3 mAb [53]. Furthermore, C_44_Mab-3 (an anti-CD44v5 mAb) [54], C_44_Mab-9 (an anti-CD44v6 mAb) [55], C_44_Mab-34 (an anti-CD44v7/8 mAb) [56], C_44_Mab-1 (an anti-CD44v9 mAb) [57], and C_44_Mab-18 (an anti-CD44v10 mAb) [58] were established. C_44_Mab-108 is an anti-CD44v4 mAb, established by peptide immunization [59]. These mAbs cover almost all variant exons and are applicable to flow cytometry, Western blotting, and immunohistochemistry.

In this study, we produced an IgG_2a_-type C_44_Mab-18 (C_44_Mab-18-mG_2a_) and investigated the antitumor efficacy against CHO/CD44v3–10 and oral SCC (OSCC) xenografts by comparing anti-pan-CD44 mAb, C_44_Mab-46-mG_2a_.

## 2. Results

### 2.1. Flow Cytometric Analysis against CHO/CD44v3–10, HSC-2, and SAS Cells Using C_44_Mab-18-mG_2a_ and C_44_Mab-46-mG_2a_

We previously established an anti-CD44v10 mAb (C_44_Mab-18, IgM, kappa) and showed the availability for flow cytometry, Western blotting, and immunohistochemistry against OSCC tissues [58]. In the present study, we cloned the V_H_ cDNA of C_44_Mab-18 and combined it with the C_H_ cDNA of mouse IgG_2a_. We also cloned the V_L_ cDNA of C_44_Mab-18 and combined it with the C_L_ cDNA of the mouse kappa light chain. Finally, a class-switched C_44_Mab-18 (C_44_Mab-18-mG_2a_) was produced (Figure 1A). We also used mouse IgG_2a_-type C_44_Mab-46 (C_44_Mab-46-mG_2a_) as an anti-pan CD44 mAb [60]. In reduced conditions, we confirm the purity of original and recombinant mAbs by SDS-PAGE (Appendix A). We next confirmed the reactivity of C_44_Mab-18-mG_2a_ and C_44_Mab-46-mG_2a_ using CHO/CD44v3–10 and CHO/CD44s cells. Both C_44_Mab-18-mG_2a_ and C_44_Mab-46-mG_2a_ detected CHO/CD44v3–10 cells in a concentration-dependent manner (Figure 1B). The reactivity of C_44_Mab-18-mG_2a_ to CHO/CD44v3–10 was superior to that of C_44_Mab-46-mG_2a_ (Figure 1B). In contrast, C_44_Mab-18-mG_2a_ did not react with CHO/CD44s cells compared to C_44_Mab-46-mG_2a_ (Figure 1C). Both C_44_Mab-18-mG_2a_ and C_44_Mab-46-mG_2a_ did not respond with CHO-K1 cells (Appendix A). These results confirmed that C_44_Mab-18-mG_2a_ recognizes only CHO/CD44v3–10 and that C_44_Mab-46-mG_2a_ recognizes both CHO/CD44v3–10 and CHO/CD44s.

We next investigated the reactivity of C_44_Mab-18-mG_2a_ and C_44_Mab-46-mG_2a_ against endogenous CD44-expressing OSCC cell lines, HSC-2 and SAS. Both C_44_Mab-18-mG_2a_ and C_44_Mab-46-mG_2a_ reacted with HSC-2 (Figure 2A) and SAS (Figure 2B) cells in a concentration-dependent manner. The reactivity of C_44_Mab-18-mG_2a_ to these cells was superior to that of C_44_Mab-46-mG_2a_ (Figure 2). We also used PMab-231 as a control mouse IgG_2a,_ which did not react with CHO/CD44v3–10, HSC-2, and SAS cells (Appendix A).

### 2.2. Induction of ADCC and CDC by C_44_Mab-18-mG_2a_ and C_44_Mab-46-mG_2a_ against CHO/CD44v3–10 Cells

Next, we performed *in vitro* ADCC and CDC assays to examine the cytotoxicity of C_44_Mab-18-mG_2a_ and C_44_Mab-46-mG_2a_ against CHO/CD44v3–10 cells. As shown in Figure 3A, C_44_Mab-18-mG_2a_ significantly exerted ADCC against CHO/CD44v3–10 cells in the presence of mouse splenocytes (30% cytotoxicity, *p* < 0.01) compared with control mouse IgG_2a_ (PMab-231, 11% cytotoxicity). Although a tendency of increased ADCC compared with the control was observed by C_44_Mab-46-mG_2a_, the difference was not significant.

In contrast, C_44_Mab-46-mG_2a_ significantly exerted CDC against CHO/CD44v3–10 cells in the presence of complements (27% cytotoxicity, *p* < 0.05) compared with the control (12% cytotoxicity). We did not observe a significant difference in CDC between C_44_Mab-18-mG_2a_ and the control (Figure 3B)

These results indicated that C_44_Mab-18-mG_2a_ and C_44_Mab-46-mG_2a_ possess different properties for exerting ADCC and CDC against CHO/CD44v3–10 cells.

### 2.3. Antitumor Effects of C_44_Mab-18-mG_2a_ and C_44_Mab-46-mG_2a_ in the Mouse Xenografts of CHO/CD44v3–10

We next evaluated the *in vivo* antitumor effect of C_44_Mab-18-mG_2a_ and C_44_Mab-46-mG_2a_ against CHO/CD44v3–10 xenograft tumors inoculated in nude mice. C_44_Mab-18-mG_2a_, C_44_Mab-46-mG_2a_, or control mouse IgG_2a_ (100 µg/mouse) were intraperitoneally injected into mice on days 7, 14, and 21 following the inoculation. The tumor volume was measured on days 7, 14, 17, 21, 24, and 28. The C_44_Mab-18-mG_2a_ and C_44_Mab-46-mG_2a_ administration resulted in a significant reduction in tumor volume on days 17 (*p* < 0.05 in C_44_Mab-18-mG_2a_), 21 (*p* < 0.01), 24 (*p* < 0.01), and 28 (*p* < 0.01) compared with that of the control mouse IgG_2a_ (Figure 4A). The C_44_Mab-18-mG_2a_ and C_44_Mab-46-mG_2a_ administration resulted in 56% and 50% reductions in tumor volume compared with the control mouse IgG_2a_ on day 28, respectively.

The weight of CHO/CD44v3–10 tumors treated with C_44_Mab-18-mG_2a_ was significantly lower than those treated with the control mouse IgG_2a_ (Figure 4B,C, 62% reduction; *p* < 0.05). Although a 53% reduction in tumor weight was observed by C_44_Mab-46-mG_2a_ treatment, it is not statistically different compared to the control (*p* = 0.0626). The loss of body weight was not observed in the CHO/CD44v3–10 tumor-implanted mice during the treatments (Figure 4D,E).

### 2.4. Induction of ADCC and CDC by C_44_Mab-18-mG_2a_ and C_44_Mab-46-mG_2a_ against HSC-2 and SAS Cells

Next, we performed *in vitro* ADCC and CDC assays to examine the cytotoxicity of C_44_Mab-18-mG_2a_ and C_44_Mab-46-mG_2a_ against HSC-2 and SAS cells. Both C_44_Mab-18-mG_2a_ and C_44_Mab-46-mG_2a_ significantly exerted ADCC against HSC-2 cells [Figure 5A, 17% (*p* < 0.01) and 13% (*p* < 0.05) cytotoxicity, respectively] and SAS cells [Figure 5C, 19% (*p* < 0.01) and 14% (*p* < 0.05) cytotoxicity, respectively] compared with the control.

Both C_44_Mab-18-mG_2a_ and C_44_Mab-46-mG_2a_ significantly exerted CDC against HSC-2 cells [Figure 5B, 27% (*p* < 0.05) and 40% (*p* < 0.01) cytotoxicity, respectively] compared with the control. In SAS cells, C_44_Mab-46-mG_2a_ significantly exerted CDC [Figure 5D, 26% (*p* < 0.05) cytotoxicity] compared with the control. We could not observe the significant difference of CDC against SAS cells between C_44_Mab-18-mG_2a_ and the control (Figure 5D).

### 2.5. Antitumor Effects of C_44_Mab-18-mG_2a_ and C_44_Mab-46-mG_2a_ in the Mouse Xenografts of HSC-2 and SAS

In the HSC-2 and SAS xenograft models, C_44_Mab-18-mG_2a_, C_44_Mab-46-mG_2a_, or control mouse IgG_2a_ (500 µg/mouse) were intraperitoneally administrated into mice on days 7 and 14, following the inoculation. The tumor volume was measured on days 7, 14, and 17. The C_44_Mab-18-mG_2a_ and C_44_Mab-46-mG_2a_ administration resulted in a significant reduction in both tumor volume on day 14 (*p* < 0.01) and day 17 (*p* < 0.01) compared with that of the control mouse IgG_2a_ (Figure 6A,B). The C_44_Mab-18-mG_2a_ and C_44_Mab-46-mG_2a_ administration resulted in 43% and 36% reduction in HSC-2 tumor volume, respectively, compared with that treated with the control mouse IgG_2a_ on day 17. The C_44_Mab-18-mG_2a_ and C_44_Mab-46-mG_2a_ administration also resulted in a 33% and 32% reduction in SAS tumor volume, respectively, compared with that treated with the control mouse IgG_2a_ on day 17.

The weight of HSC-2 tumors treated with C_44_Mab-18-mG_2a_ and C_44_Mab-46-mG_2a_ was significantly lower than that treated with the control mouse IgG_2a_ [Figure 6C, 40% (*p* < 0.01) and 30% (*p* < 0.05) reduction, respectively]. The weight of SAS tumors treated with C_44_Mab-18-mG_2a_ and C_44_Mab-46-mG_2a_ was significantly lower than that treated with the control mouse IgG_2a_ [Figure 6D, 23% (*p* < 0.01) and 19% (*p* < 0.05) reduction, respectively]. The loss of body weight was not observed in the HSC-2 and SAS tumor-implanted mice during the treatments (Figure 6E,F).

## 3. Discussion

Among CD44v, v10-containing isoforms include the most abundant CD44v, such as CD44v3–10, CD44v6–10, and CD44v8–10 [2,61]. Therefore, anti-CD44v10 mAbs including C_44_Mab-18-mG_2a_ can target the broad range of CD44v-expressing tumor cells. In this study, we evaluated the antitumor activities against CD44-positive cells compared with an anti-pan-CD44 mAb, C_44_Mab-46-mG_2a_. C_44_Mab-18-mG_2a_ exhibited the higher reactivity to CHO/CD44v3–10 and OSCC cells compared with C_44_Mab-46-mG_2a_ (Figure 1 and Figure 2). C_44_Mab-18-mG_2a_ exhibited a superior ADCC against CHO/CD44v3–10 (Figure 3) and OSCC (Figure 5) cells. In contrast, C_44_Mab-46-mG_2a_ showed a superior CDC against those cells. Furthermore, C_44_Mab-18-mG_2a_ or C_44_Mab-46-mG_2a_ similarly inhibited CHO/CD44v3–10 and OSCC xenograft growth compared with the control mouse IgG_2a_ (Figure 4 and Figure 6). These results indicate that C_44_Mab-18-mG_2a_ could be a promising therapeutic regimen for CD44v10-positive tumors.

OSCC arises from the oral cavity and is a type of head and neck squamous cell carcinoma (HNSCC). HNSCC has been revealed to be associated with alcohol, smoking, and human papillomavirus types 16 and 18 infection [62]. In the Pan-Cancer Atlas, HNSCC has been shown as the second-highest CD44-expressing tumor type [63]. The overexpression of CD44 is associated with resistance to therapy and unfavorable outcomes [64,65,66]. Furthermore, CD44-high cancer stem cells (CSCs) from HNSCC exhibited increased migration, invasiveness, and stemness [67]. In immunodeficient mice, the CD44-high CSCs could form more lung metastatic foci than CD44-low cells [68]. Therefore, CD44 is considered an important target for mAb therapies. Since C_44_Mab-18 is available to immunohistochemistry [58], C_44_Mab-18 could be used for diagnosis and therapy of OSCC.

In CHO/CD44v3–10 cells, C_44_Mab-18-mG_2a_ and C_44_Mab-46-mG_2a_ recognized a common target but mainly exerted ADCC and CDC activity, respectively (Figure 3). The epitope of C_44_Mab-46 was previously determined as the _174-_TDDDV_-178_ sequence in the constant exon 5-encoded region [51], which is relatively apart from transmembrane domain compared to variant exon 10-encoded region recognized by C_44_Mab-18 [58]. To activate the classical pathway of complement, an ordered hexamer formation of IgG mAb is required to bind to the hexavalent complement C1q [69,70]. The structure of the C_44_Mab-46-mG_2a_–CD44v3–10 complex may provide the appropriate space to form the hexameric structure of the mAb–C1q complex and recruit the pore-forming membrane attack complex to exert CDC. In contrast, C_44_Mab-18-mG_2a_ showed a higher reactivity to CHO/CD44v3–10 compared with C_44_Mab-46-mG_2a_ in flow cytometry (Figure 1). The difference in the reactivity and the epitope would influence the ADCC activity. Further studies are required to reveal the relationship among the affinity of mAb, epitope, and ADCC activity.

The limitation of this study is that both C_44_Mab-46-mG_2a_ and C_44_Mab-18-mG_2a_ are effective in human xenograft models of nude mice. To apply human tumor therapy, a class switch to human IgG_1_ is essential. We previously generated humanized IgG_1_ mAbs and evaluated the antitumor activity with injections of human NK cells [71,72]. We will produce humanized mAbs from C_44_Mab-46 and C_44_Mab-18 and evaluate the ADCC and CDC in the presence of human NK cells and complement, respectively. Furthermore, the antitumor effects should be investigated with injections of human NK cells.

Near-infrared photoimmunotherapy (NIR-PIT) uses a targeted mAb conjugated with a photoactivatable dye such as IRDye700DX (IR700) [73,74,75,76]. When the mAb binds to the antigen-expressed target cells, IR700 induces plasma membrane rupture and immunogenic cell death by NIR-light exposure. Preclinical studies of anti-pan-CD44 mAb-based NIR-PIT (IM7-IR700) were conducted. In the syngeneic mouse model of OSCC, IM7-IR700 administration and the NIR light exposure to OSCC tumors resulted in a significant reduction but failed to induce durable antitumor responses [77]. Because IM7 is a pan-CD44 mAb, IM7 might target not only tumor cells but also CD44s-positive immune cells which are involved in the antitumor immunity. The expression of CD44v is low in hematopoietic cells compared with CD44s [2]. We previously showed that C_44_Mab-18 can distinguish tumor cells from stromal tissues in immunohistochemistry. In contrast, C_44_Mab-46 stained both tumor and stromal tissue including fibroblasts and leukocytes [58]. Therefore, anti-CD44v10 mAbs such as C_44_Mab-18 might be a promising mAb for NIR-PIT without affecting the host immune cells in the tumor microenvironment.

Since both CD44s and CD44v are expressed in normal cells, there is a concern about adverse effects due to the recognition of normal cells by mAbs. In fact, clinical trials of the anti-CD44v6 mAb–ADC to advanced solid tumors were discontinued because of the skin toxicities [32,37]. Therefore, cancer-specific antibodies are desired to reduce the adverse effects. A cancer-specific anti-CD44v6 mAb (clone 4C8) recognizes aberrantly *O*-glycosylated Tn (GalNAcα1-*O*-Ser/Thr) antigen in the variant exon 6-encoded region. The 4C8 mAb was further developed for chimeric antigen receptor (CAR)-T cells, which exhibited target-specific in vitro cytotoxicity and significant tumor regression in vivo [78]. We have developed cancer-specific mAbs (CasMabs) against various tumor antigens, including HER2 (clones H_2_Mab-214 [79] and H_2_Mab-250 [80]), and reported the antitumor effect in mouse xenograft models using recombinant mouse IgG_2a_ or human IgG_1_ mAbs [71,72]. These anti-HER2 mAbs were screened by the reactivity to cancer and normal cells in flow cytometry. H_2_Mab-214 was revealed to recognize a locally misfolded structure in the Cys-rich HER2 extracellular domain 4, which usually forms a β-sheet [79]. H_2_Mab-250 also shows a specific reactivity against HER2-positive tumor cells, which has been developed as CAR-T-cell therapy. The phase I study has been conducted in the US (NCT06241456). We have developed CasMabs against CD44s or CD44v by comparing the reactivity against tumor and normal cells. The anti-CD44 CasMabs could contribute to developing novel modalities such as ADCs and CAR-T cells.

## 4. Materials and Methods

### 4.1. Cell Lines and Cell Culture

CHO-K1 was obtained from the American Type Culture Collection (ATCC, Manassas, VA, USA). OSCC cell lines HSC-2 and SAS were obtained from the Japanese Collection of Research Bioresources (Osaka, Japan). CHO/CD44s and CHO/CD44v3–10 were previously established by transfecting pCAG-Ble/PA16-CD44s and pCAG-Ble/PA16-CD44v3–10 into CHO-K1 cells using a Neon transfection system (Thermo Fisher Scientific, Inc., Waltham, MA, USA) [50,58].

HSC-2 and SAS were cultured in Dulbecco’s Modified Eagle Medium (DMEM, Nacalai Tesque, Inc., Kyoto, Japan) containing 10% (*v*/*v*) heat-inactivated fetal bovine serum (FBS, Thermo Fisher Scientific Inc.), 100 U/mL of penicillin (Nacalai Tesque, Inc.), 100 μg/mL streptomycin (Nacalai Tesque, Inc.), and 0.25 μg/mL amphotericin B (Nacalai Tesque, Inc.).

CHO/CD44v3–10 was cultured in Roswell Park Memorial Institute (RPMI)-1640 medium (Nacalai Tesque, Inc.) supplemented with 10% (*v*/*v*) FBS, antibiotics as mentioned above, and 5 mg/mL Zeocin (InvivoGen, San Diego, CA, USA). All cells were grown in a humidified incubator at 37 °C with 5% CO_2_.

### 4.2. Antibodies

An anti-pan-CD44 mAb (C_44_Mab-46) and an anti-CD44v10 mAb (C_44_Mab-18) were previously established [50,58]. A recombinant mouse IgG_2a_-type mAb, C_44_Mab-46-mG_2a,_ was generated previously [60]. To generate a recombinant mouse IgG_2a_-type mAb from C_44_Mab-18 (IgM, kappa), V_H_ cDNAs of C_44_Mab-18 and C_H_ of mouse IgG_2a_ were cloned into the pCAG-Ble vector (FUJIFILM Wako Pure Chemical Corporation, Osaka, Japan). V_L_ and mouse kappa light chain (C_L_) cDNA of C_44_Mab-18 was also cloned into the pCAG-Neo vector (FUJIFILM Wako Pure Chemical Corporation). Using the ExpiCHO Expression System (Thermo Fisher Scientific, Inc.), the vectors were transfected into BINDS-09 cells (http://www.med-tohoku-antibody.com/topics/001_paper_cell.htm, accessed on 23 August 2024) and the supernatants were collected. C_44_Mab-18-mG_2a_ were purified using Ab-Capcher (ProteNova Co., Ltd., Kagawa, Japan). PMab-231 (a control mouse IgG_2a_) was previously described [80].

### 4.3. Flow Cytometry

CHO/CD44v3–10, HSC-2, and SAS were obtained using 1 mM ethylenediamine tetraacetic acid (EDTA; Nacalai Tesque, Inc.) and 0.25% trypsin treatment. The cells were treated with C_44_Mab-18-mG_2a_, C_44_Mab-46-mG_2a_, PMab-231, or blocking buffer [0.1% bovine serum albumin (BSA; Nacalai Tesque, Inc.) in phosphate-buffered saline (PBS)] (control) for 30 min at 4 °C. Subsequently, the cells were incubated in Alexa Fluor 488-conjugated anti-mouse IgG (1:2000; Cell Signaling Technology, Inc., Danvers, MA, USA) for 30 min at 4 °C. Fluorescence data were collected using the SA3800 Cell Analyzer (Sony Corp., Tokyo, Japan) and analyzed using SA3800 software ver. 2.05 (Sony Corp.) [50].

### 4.4. ADCC

Animal studies for ADCC were approved by the Institutional Committee for Experiments of the Institute of Microbial Chemistry (permit no. 2024-038). The ADCC activity of C_44_Mab-18-mG_2a_ and C_44_Mab-46-mG_2a_ was measured as follows. The target cells (CHO/CD44v3–10, HSC-2, and SAS) were labeled with 10 µg/mL Calcein AM (Thermo Fisher Scientific, Inc.) and plated in 96-well plates (1 × 10^4^ cells/well). The calcein-labeled target cells were mixed with the effector splenocyte (effector to target ratio, 50:1) from female BALB/c nude mice (Jackson Laboratory Japan, Inc., Kanagawa, Japan) with 100 μg/mL of C_44_Mab-18-mG_2a_, C_44_Mab-46-mG_2a_, or control mouse IgG_2a_ (PMab-231). After a 4 h incubation at 37 °C, the calcein release into the medium was measured using a microplate reader (Power Scan HT; BioTek Instruments, Inc., Winooski, VT, USA).

The cytolyticity (% lysis) was determined: % lysis is calculated as (E − S)/(M − S) × 100, where “E” indicates the fluorescence in effector and target cell cultures, “S” means the spontaneous fluorescence of only target cells, and “M” indicates the maximum fluorescence after treatment with a lysis buffer [10 mM Tris-HCl (pH 7.4), 10 mM EDTA, and 0.5% Triton X-100] [49].

### 4.5. CDC

The calcein-labeled target cells were mixed with rabbit complement (final concentration 15%, Low-Tox-M Rabbit Complement; Cedarlane Laboratories, Hornby, ON, Canada) and 100 μg/mL of C_44_Mab-18-mG_2a_, C_44_Mab-46-mG_2a_, or control mouse IgG_2a_ (PMab-231). After a 4 h incubation at 37 °C, the calcein release into the medium was measured as described previously [60].

### 4.6. Antitumor Activity of C_44_Mab-18-mG_2a_ and C_44_Mab-46-mG_2a_ in Xenografts of CHO/CD44v3–10, HSC-2, and SAS

The animal study protocol was approved (approval nos. 2024-013) by the Institutional Committee for Experiments of the Institute of Microbial Chemistry (Numazu, Japan). The animal study was performed as described previously [49]. CHO/CD44v3–10, HSC-2, or SAS cells (5 × 10^6^ cells) suspended with BD Matrigel Matrix Growth Factor Reduced (BD Biosciences, Franklin Lakes, NJ, USA) were inoculated into the left flank of female BALB/c nude mice subcutaneously. On day 7 after the inoculation, 100 μg of C_44_Mab-18-mG_2a_ (n = 8), C_44_Mab-46-mG_2a_ (n = 8), or control mouse IgG_2a_ (PMab-231) (n = 8) in 100 μL PBS was injected intraperitoneally. Additional antibody injections were performed on days 14 and 21. The tumor volume was measured on the indicated days.

In the HSC-2 or SAS xenograft experiment, 500 μg of C_44_Mab-18-mG_2a_ (n = 8), C_44_Mab-46-mG_2a_ (n = 8), or control mouse IgG_2a_ (PMab-231) (n = 8) in 100 μL PBS was injected intraperitoneally on days 7 and 14 after the inoculation. The tumor volume was measured on the indicated days. The xenograft tumors were carefully removed from the sacrificed mice and weighed immediately.

The tumor volume was calculated using the following formula: Volume = W^2^ × L/2, where W is the short diameter and L is the long diameter. All data are expressed as the mean ± standard error of the mean (SEM). In tumor weight measurement, one-way ANOVA with Tukey’s multiple comparisons test was conducted. Two-way ANOVA with Tukey’s multiple comparisons test was utilized for tumor volume and mice weight. GraphPad Prism 6 (GraphPad Software, Inc., La Jolla, CA, USA) was used for all calculations. *p* < 0.05 was considered to indicate a statistically significant difference.

## Figures and Tables

**Figure 1 ijms-25-09190-f001:**
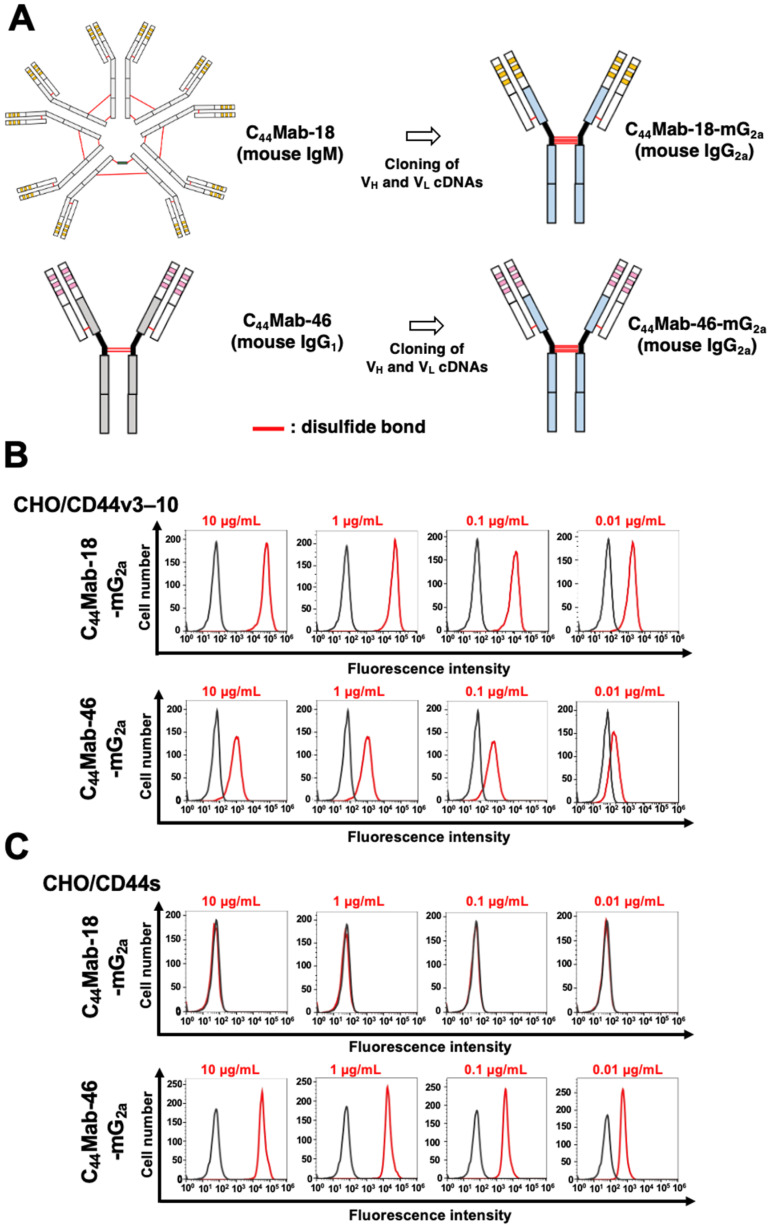
Flow cytometry using C_44_Mab-18-mG_2a_ and C_44_Mab-46-mG_2a_. (**A**) Class-switched mouse IgG_2a_ mAbs, C_44_Mab-18-mG_2a,_ and C_44_Mab-46-mG_2a_ were generated from C_44_Mab-18 (mouse IgM) and C_44_Mab-46 (mouse IgG_1_), respectively. CHO/CD44v3–10 (**B**) and CHO/CD44s (**C**) cells were treated with buffer control (black) or 10–0.01 µg/mL of C_44_Mab-18-mG_2a_ and C_44_Mab-46-mG_2a_ (red). The cells were further treated with Alexa Fluor 488-conjugated anti-mouse IgG. Fluorescence data were analyzed using the SA3800 Cell Analyzer.

**Figure 2 ijms-25-09190-f002:**
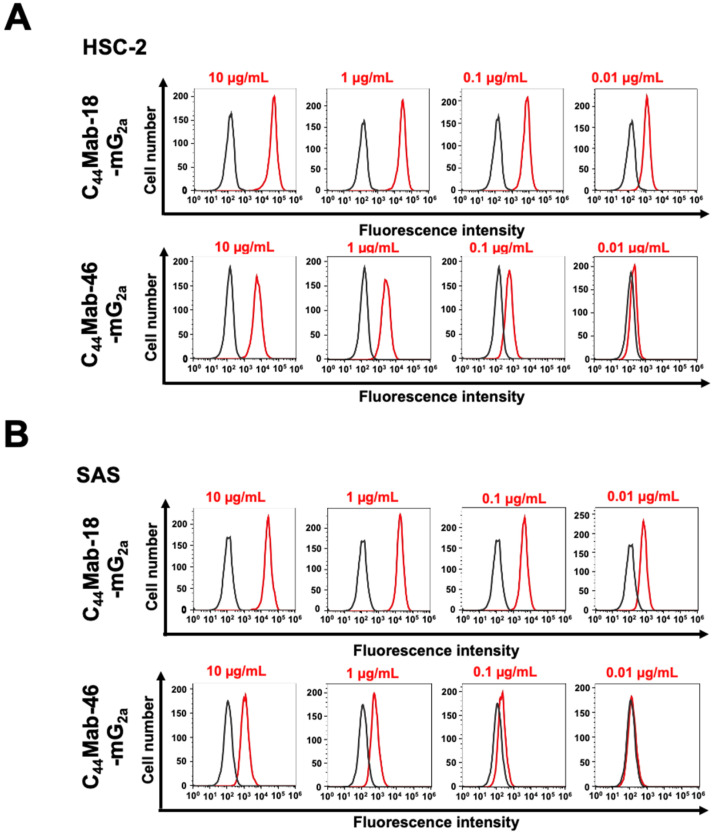
Flow cytometry using C_44_Mab-18-mG_2a_ and C_44_Mab-46-mG_2a_ against oral squamous cell carcinoma (OSCC) cell lines. HSC-2 (**A**) and SAS (**B**) cells were treated with buffer control (black) or 10–0.01 µg/mL of C_44_Mab-18-mG_2a_ and C_44_Mab-46-mG_2a_ (red). The cells were further treated with Alexa Fluor 488-conjugated anti-mouse IgG. Fluorescence data were analyzed using the SA3800 Cell Analyzer.

**Figure 3 ijms-25-09190-f003:**
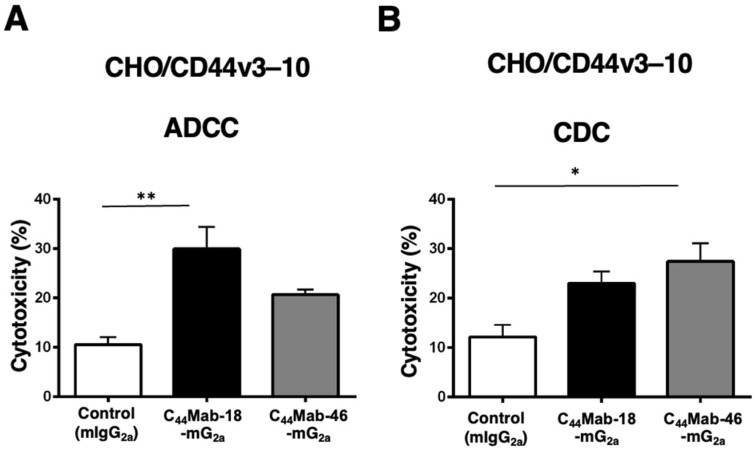
Evaluation of ADCC and CDC activity of C_44_Mab-18-mG_2a_ and C_44_Mab-46-mG_2a_ against CHO/CD44v3–10 cells. The ADCC (**A**) and CDC (**B**) induced by C_44_Mab-18-mG_2a_, C_44_Mab-46-mG_2a_, or control mouse IgG_2a_ (mIgG_2a_, PMab-231) against CHO/CD44v3–10 cells. Values are shown as mean ± SEM. Asterisks indicate statistical significance (** *p* < 0.01 and * *p* < 0.05; one-way ANOVA with Tukey’s multiple comparisons test).

**Figure 4 ijms-25-09190-f004:**
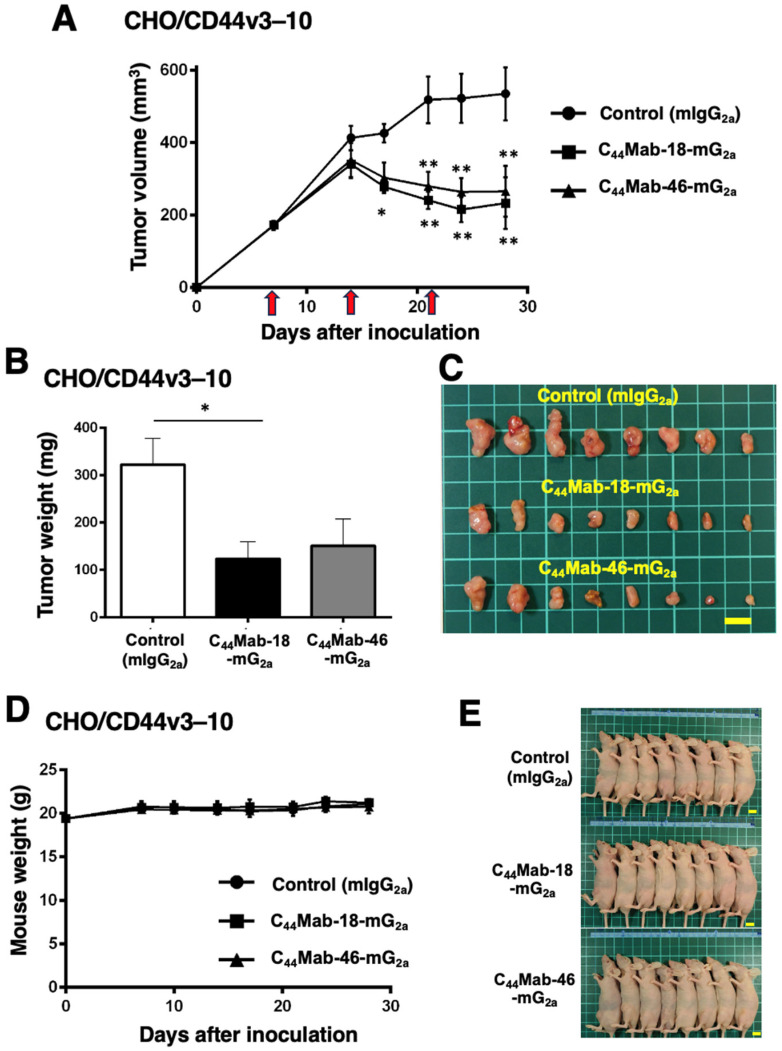
Antitumor activity of C_44_Mab-18-mG_2a_ and C_44_Mab-46-mG_2a_ against CHO/CD44v3–10 xenograft. (**A**) Tumor volume of CHO/CD44v3–10 xenograft. CHO/CD44v3–10 cells (5 × 10^6^ cells) were injected into mice subcutaneously. On days 7, 14, and 21, 100 μg of C_44_Mab-18-mG_2a_ (n = 8), C_44_Mab-46-mG_2a_ (n = 8), or control mouse IgG_2a_ (mIgG_2a_, PMab-231) (n = 8) was injected into mice intraperitoneally (arrows). The tumor volume was measured on days 7, 14, 17, 21, 24, and 28 following the inoculation. Values are presented as the mean ± SEM. * *p* < 0.05 and ** *p* < 0.01 (two-way ANOVA with Tukey’s multiple comparisons test). The weight (**B**) and appearance (**C**) of excised CHO/CD44v3–10 xenografts on day 28. Values are presented as the mean ± SEM. * *p* < 0.05 (one-way ANOVA with Tukey’s multiple comparisons test). The body weight (**D**) and appearance (**E**) of CHO/CD44v3–10 xenograft-bearing mice treated with C_44_Mab-18-mG_2a_, C_44_Mab-46-mG_2a_, or control mIgG_2a_. Scale bar, 1 cm.

**Figure 5 ijms-25-09190-f005:**
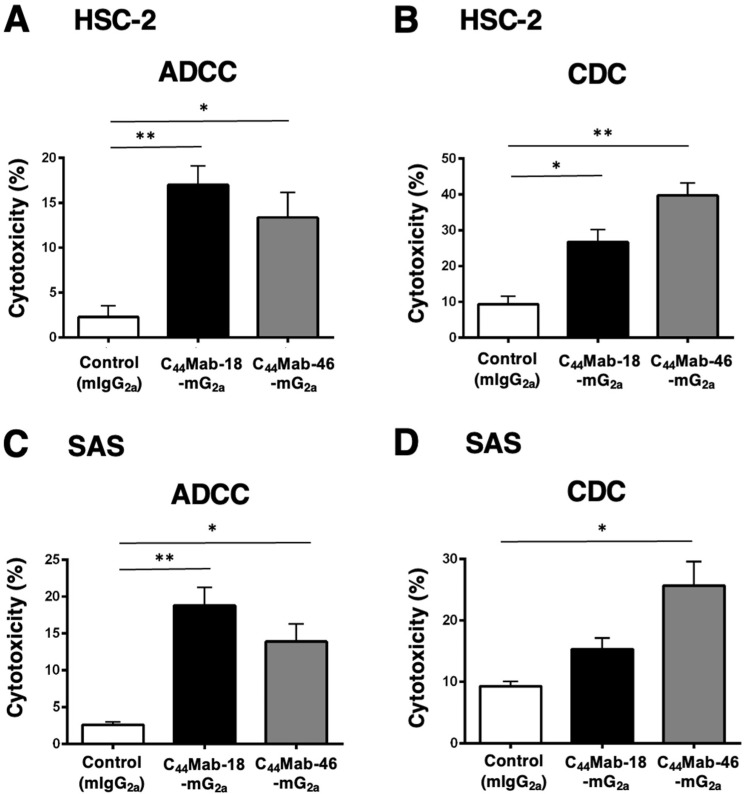
Evaluation of ADCC and CDC activity of C_44_Mab-18-mG_2a_ and C_44_Mab-46-mG_2a_ against HSC-2 and SAS cells. The ADCC (**A**,**C**) and CDC (**B**,**D**) induced by C_44_Mab-18-mG_2a_, C_44_Mab-46-mG_2a_, or control mouse IgG_2a_ (mIgG_2a_, PMab-231) against HSC-2 (**A**,**B**) and SAS (**C**,**D**) cells. Values are shown as mean ± SEM. Asterisks indicate statistical significance (** *p* < 0.01 and * *p* < 0.05; one-way ANOVA with Tukey’s multiple comparisons test).

**Figure 6 ijms-25-09190-f006:**
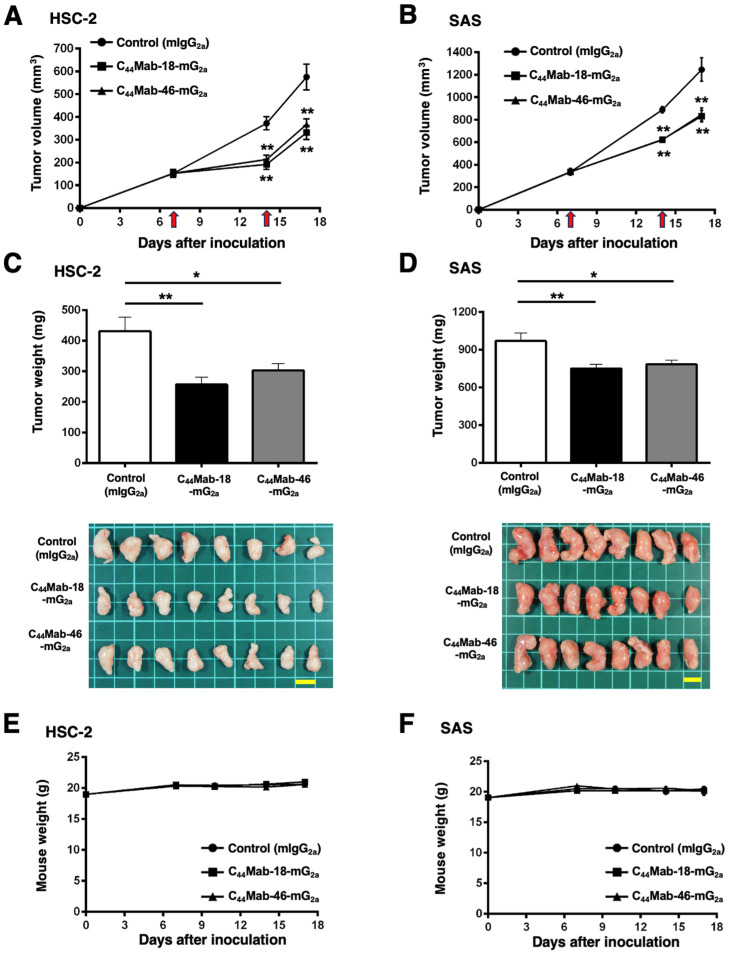
Antitumor activity of C_44_Mab-18-mG_2a_ and C_44_Mab-46-mG_2a_ against HSC-2 and SAS xenograft. (**A**,**B**) Tumor volume in HSC-2 (**A**) and SAS (**B**) xenograft. HSC-2 and SAS cells (5 × 10^6^ cells) were injected into mice subcutaneously. On days 7 and 14, 500 μg of C_44_Mab-18-mG_2a_ (n = 8), C_44_Mab-46-mG_2a_ (n = 8), or control mouse IgG_2a_ (mIgG_2a_, PMab-231) (n = 8) was injected into mice intraperitoneally (arrows). The tumor volume was measured on days 7, 14, and 17 following the inoculation. Values are presented as the mean ± SEM. ** *p* < 0.01 (two-way ANOVA with Tukey’s multiple comparisons test). (**C**,**D**) The weight and appearance of the excised HSC-2 (**C**) and SAS (**D**) xenografts on day 17. Values are presented as the mean ± SEM. * *p* < 0.05 and ** *p* < 0.01 (one-way ANOVA with Tukey’s multiple comparisons test). (**E**,**F**) The body weight of HSC-2 (**E**) and SAS (**F**) xenograft-bearing mice treated with C_44_Mab-18-mG_2a_, C_44_Mab-46-mG_2a_, or control mIgG_2a_. Scale bar, 1 cm.

## Data Availability

The data presented in this study are available in the article and Appendix A.

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
