# Peer review of "Anti-CD44 Variant 10 Monoclonal Antibody Exerts Antitumor Activity in Mouse Xenograft Models of Oral Squamous Cell Carcinomas"

_ijms, 2024, doi:10.3390/ijms25179190_

Round 1

Reviewer 1 Report

Comments and Suggestions for Authors

The authors extensively evaluated the anti tumor activity of anti-CD44 variant 10 monoclonal antibody against mouse xenograft models of oral squamous cell carcinomas. Multiple techniques and approaches have been used to study the antiproliferative effect generating a sound document and well-written. The research was considered well-qualified. The paper's results are organized well, Authors also are well familiar with the ethical consideration and the protocol received ethical approval from institutional committee  

However, there are several points to be improved as mentioned below.      

·         keywords. Remove all indicated keywords that appear in the title

·         In the animal study, why authors used 8 animals for each group?

·         How the authors calculated the sample size for animal study?

·         line 360-362: no need to mention this here it is already mentioned in the statistical analysis section it is a repetition, also 368-369 should be removed.

·         The methods in section; materials and methods, have no references. Authors should cite the reference of the reported method and if any modification was made, they should state this clearly 

·         The method of scarification was not mentioned

·         It is not mentioned whether the animals were anesthetized before cervical dislocation, please provide the type and dose of anesthesia used 

·         The extensive self-citation is seen in the reference's list eg: 28-40, 47-51 while some are relevant, multiple are not necessarily.

Author Response

The authors extensively evaluated the antitumor activity of anti-CD44 variant 10 monoclonal antibody against mouse xenograft models of oral squamous cell carcinomas. Multiple techniques and approaches have been used to study the antiproliferative effect generating a sound document and well-written. The research was considered well-qualified. The paper's results are organized well, Authors also are well familiar with the ethical consideration and the protocol received ethical approval from institutional committee.

However, there are several points to be improved as mentioned below.

  • keywords. Remove all indicated keywords that appear in the title

-> We removed and added other key words.

  • In the animal study, why authors used 8 animals for each group?

-> We used 8 animals for each group to ensure a significant difference. We have been performed animal experiments (ref 60, 77, 79, and 80) as above.

  • How the authors calculated the sample size for animal study?

-> We added a method to calculate the size of xenograft, as follows.

(Page 12, line 410)

The tumor volume was calculated using the formula: Volume = W2 x L/2, where W is the short diameter and L is the long diameter.

  • line 360-362: no need to mention this here it is already mentioned in the statistical analysis section it is a repetition, also 368-369 should be removed.

-> We removed both.

  • The methods in section; materials and methods, have no references. Authors should cite the reference of the reported method and if any modification was made, they should state this clearly

-> We cited the references.

  • The method of scarification was not mentioned

-> Does “scarification” mean “sacrifice”?

We sacrificed mice by cervical dislocation.

  • It is not mentioned whether the animals were anesthetized before cervical dislocation, please provide the type and dose of anesthesia used.

-> Anesthesia was not used before cervical dislocation.

  • The extensive self-citation is seen in the reference's list eg: 28-40, 47-51 while some are relevant, multiple are not necessarily.

-> We reduced the self-citation ratio according to the guideline.

Reviewer 2 Report

Comments and Suggestions for Authors

The study conducted by Ishikawa et al. explores the efficacy of a new monoclonal antibody, C44Mab-18-mG2a, in specifically targeting tumors that express CD44v10, in mouse xenograft models of oral squamous cell carcinomas. The study shows encouraging anticancer effects in both under invitro as well as invivo models, indicating that C44Mab-18-mG2a has the potential to be a beneficial treatment choice for malignancies that exhibit CD44v10 expression.

1. The study primarily aimed to show the anticancer effects of C44Mab-18-mG2a, but does not extensively investigate the underlying mechanisms. Additional research is required to clarify the mechanism by which C44Mab-18-mG2a binds to CD44v10 and induces tumor cell death. Gaining a comprehensive understanding of the precise mechanisms involved could facilitate the identification of potential biomarkers that indicate a response and enhance the optimization of treatment regimens.

2.Comparison with Other Anti-CD44 Antibodies: The study juxtaposes C44Mab-18-mG2a with C44Mab-46-mG2a, an antibody that targets the whole CD44 protein. Nevertheless, it would be enlightening to assess the effectiveness and safety of C44Mab-18-mG2a in comparison to other antibodies that target CD44v or alternative therapeutic strategies that aim at CD44v10-positive malignancies.

3.Due to the complex characteristics of cancer, it would be advantageous to investigate the potential of C44Mab-18-mG2a in conjunction with other treatments, such as chemotherapy or immunotherapy. These investigations have the potential to uncover synergistic effects and improve the overall therapeutic response.

4.Is it possible to synergistically boost the anti-tumor activity of C44Mab-18-mG2a by combining it with other therapeutic drugs, while simultaneously reducing potential side effects?

5.Which molecular pathways does C44Mab-18-mG2a specifically target to provide its anti-tumor actions in oral squamous cell carcinoma?

Comments on the Quality of English Language

NA

Author Response

The study conducted by Ishikawa et al. explores the efficacy of a new monoclonal antibody, C44Mab-18-mG2a, in specifically targeting tumors that express CD44v10, in mouse xenograft models of oral squamous cell carcinomas. The study shows encouraging anticancer effects in both under invitro as well as invivo models, indicating that C44Mab-18-mG2a has the potential to be a beneficial treatment choice for malignancies that exhibit CD44v10 expression.

  1. The study primarily aimed to show the anticancer effects of C44Mab-18-mG2a, but does not extensively investigate the underlying mechanisms. Additional research is required to clarify the mechanism by which C44Mab-18-mG2a binds to CD44v10 and induces tumor cell death.

-> In mAb therapy for tumors, ADCC and CDC are important mechanism to induce tumor cell death. As we showed in Figure 3 and 5, both C44Mab-18-mG2a and C44Mab-46-mG2a exerted the ADCC and CDC against CHO/CD44v3-10, HSC-2, and SAS cells.

Gaining a comprehensive understanding of the precise mechanisms involved could facilitate the identification of potential biomarkers that indicate a response and enhance the optimization of treatment regimens.

-> In mAb therapy for tumors, the expression of antigen is the most important biomarker to select the patients. For instance, trastuzumab is administered in patients with HER2-overexpressed tumors, which are defined by solid and complete membranous staining of more than 10% of cells in immunohistochemistry (IHC 3+). Since C44Mab-18 is available to IHC, it could be used to define the CD44v10-overexpressed tumors.

We added the discussion about the importance of CD44 in OSCC, as follows.

(Page 10, line 275)

OSCC arises from the oral cavity and is a type of head and neck squamous cell carcinoma (HNSCC). HNSCC has been revealed to associate with alcohol, smoking, and human papillomavirus types 16 and 18 infection [62]. In the Pan-Cancer Atlas, HNSCC has been shown as the second-highest CD44-expressing tumor type [63]. The overexpression of CD44 is associated with resistance to therapy and unfavorable outcomes [64-66]. Furthermore, CD44-high cancer stem cells (CSCs) from HNSCC exhibited the increased migration, invasiveness, and stemness [67]. In immunodeficient mice, the CD44-high CSCs could form more lung metastatic foci than CD44-low cells [68]. Therefore, CD44 is considered an important target for mAb therapies. Since C44Mab-18 is available to immunohistochemistry [58], C44Mab-18 could be used for diagnosis and therapy of OSCC.

2.Comparison with Other Anti-CD44 Antibodies: The study juxtaposes C44Mab-18-mG2a with C44Mab-46-mG2a, an antibody that targets the whole CD44 protein. Nevertheless, it would be enlightening to assess the effectiveness and safety of C44Mab-18-mG2a in comparison to other antibodies that target CD44v or alternative therapeutic strategies that aim at CD44v10-positive malignancies.

-> CD44v is selectively expressed in normal epithelial tissue and the malignancies including SCC. In contrast, CD44s is broadly expressed in normal tissues and cancers. As we mentioned in discussion, v10-containing isoforms are the most abundant CD44v, such as CD44v3–10, CD44v6–10, and CD44v8–10. Therefore, we compared the antitumor effects between C44Mab-18-mG2a and C44Mab-46-mG2a. Both mAbs showed a similar antitumor effect (Figure 4 and 6). To our surprise, the contribution of ADCC and CDC was different (Figure 3 and 5). In the future study, we should evaluate the antitumor effect of each variant-specific anti-CD44v mAb with determining the ADCC and CDC.

3.Due to the complex characteristics of cancer, it would be advantageous to investigate the potential of C44Mab-18-mG2a in conjunction with other treatments, such as chemotherapy or immunotherapy. These investigations have the potential to uncover synergistic effects and improve the overall therapeutic response.

-> The combination therapy of C44Mab-18-mG2a with chemotherapy or immunotherapy is the important strategy for cancer treatment. However, the aim of study is to reveal the antitumor effect of C44Mab-18-mG2a monotherapy. Therefore, we would like to investigate the combination therapy of C44Mab-18-mG2a in the future studies.

4.Is it possible to synergistically boost the anti-tumor activity of C44Mab-18-mG2a by combining it with other therapeutic drugs, while simultaneously reducing potential side effects?

-> In our model, we cannot evaluate the side effects of C44Mab-18-mG2a because C44Mab-18-mG2a does not recognize mouse CD44v10. C44Mab-18-mG2a may potentiate antitumor activity of cisplatin, an important chemotherapeutic drug for OSCC therapy, whith reducing the side effects. As mentioned above, we would like to investigate the combination therapy of C44Mab-18-mG2a with the chemotherapy in the future studies.

5.Which molecular pathways does C44Mab-18-mG2a specifically target to provide its anti-tumor actions in oral squamous cell carcinoma?

-> As we mentioned in comment 1, ADCC and CDC are major mechanism to induce tumor cell death. As we showed in Figure 3 and 5, both C44Mab-18-mG2a and C44Mab-46-mG2a exerted the ADCC and CDC against CHO/CD44v3-10, HSC-2, and SAS cells. We have not investigated the effect of C44Mab-18-mG2a on the CD44v10-mediated signaling pathways. Since C44Mab-18 does not recognize hyaluronic acid (HA, a CD44 ligand)-binding domain, C44Mab-18 would not have the neutralization activity against HA.

Round 2

Reviewer 1 Report

Comments and Suggestions for Authors

The authors answered all the comments 

Author Response

Thank you very much.